# Spin-Noise Gradient Echoes

Victor V. Rodin[1], Stephan J. Ginthör[1], Matthias Bechmann[1], Hervé Desvaux[2], and Norbert Müller[1,3]

[1]Institute of Organic Chemistry, Johannes Kepler University Linz, Altenbergerstraße 69, 4040 Linz, Austria
[2]NIMBE, CEA, CNRS, Université Paris-Saclay, CEA/Saclay, 91191 Gif-sur-Yvette, France
[3]Faculty of Science, University of South Bohemia in České Budějovice, Branišovská 1645/31a, 370 05 České Budějovice, Czech Republic

*Correspondence to*: Norbert Müller (norbert.mueller@jku.at)

**Abstract.** Nuclear spin-noise spectroscopy in absence of radio frequency pulses was studied under the influence of pulsed field gradients (PFGs) on pure and mixed liquids. Under conditions, where the radiation-damping induced line broadening is
smaller than the gradient dependent inhomogeneous broadening, echo responses can be observed in difference spectra between experiments employing pulsed field gradient pairs of same and opposite signs. These observed "spin-noise gradient echoes" (SNGEs) were analyzed through a simple model to describe the effects of transient phenomena. Experiments performed on high resolution NMR probes demonstrate how "refocused spin noise" behaves and how it can be exploited to determine sample properties. In bulk liquids and their mixtures transverse relaxation times as well as translational diffusion constants can be
determined from SNGE spectra recorded following tailored sequences of magnetic field gradient pulses.

## 1 Introduction

Felix Bloch (Bloch, 1946) predicted nuclear spin noise (SN) more than 70 years ago as the result of incomplete cancellation of random fluctuations of spin polarization. After the first experimental observation of a weak nuclear quadrupole resonance (NQR) noise spectrum by Sleator (Sleater et al., 1985) SN has become a subject of renewed and increased interest (Guéron
and Leroy, 1989; Marion and Desvaux, 2008; McCoy and Ernst, 1989; Müller and Jerschow, 2006; Pöschko et al., 2017). In particular, it has a really appealing potential for studying nano-scale samples (Nichol et al., 2014). The intensity of the SN signal observed in experiments without any radio frequency (RF) pulses is circa 108 times smaller than the signal obtained in the case of 90° RF pulse excitation for thermally polarized $^1$H nuclear spin systems at 500 MHz at millimolar concentration (Marion and Desvaux, 2008; Nichol et al., 2014; Pöschko et al., 2017). McCoy and Ernst (McCoy and Ernst, 1989) studied
nuclear spin-noise spectra in ethanol at room temperature by co-adding thousands of 1D power spectra acquired without RF excitation. When correlated noise was distinguished from uncorrelated noise by a cross-correlation process, 2D Fourier transform NMR studies resulted in detected spin-noise spectra from a liquid sample of macroscopic size (Chandra et al., 2013). In a recent publication on the "double-block usage" processing scheme (Ginthör et al., 2018), each recorded spin-noise block was used in two independent cross correlations. With such an approach, the sensitivity of 2D spin-noise spectroscopy has been
increased significantly. Nuclear spin noise accumulated in the presence of magnetic field gradients applied in different directions was used to implement spin-noise imaging in the absence of any RF pulses applying a projection-reconstruction approach for data processing (Müller and Jerschow, 2006).

Many modern high-resolution NMR spectrometers are equipped with cryogenically cooled probe systems, which reduce electronic noise to a minimum and are therefore the preferred probes for spin noise studies (Bloom, 1957; Desvaux, 2013;
Nichol et al., 2014; Pöschko et al., 2014). However, owed to low noise electronics, SN spectra can even be obtained relatively easily using room temperature probes on samples with sufficiently large numbers of spins (order of $10^{17}$ – $10^{20}$). In spite of the relatively straightforward measurement procedures, the line shapes of spin-noise resonances can exhibit complex features. This is, because spin noise and radiation damping (RD) are virtually inseparable phenomena, giving rise to highly non-linear behavior and frequency shifts (Bloch, 1946; Bloom, 1957; McCoy and Ernst, 1989; Guéron and Leroy, 1989; Nausner et al.,
2009; Desvaux, 2013; Krishnan and Murali, 2013; Nichol et al., 2014; Ferrand et al., 2015; Pöschko et al., 2015). Long-

standing unresolved questions concerning quantitative discrepancies between experiment and the theory derived by (McCoy and Ernst, 1989) as well as (Guéron, 1991) have recently been largely reconciled (Ferrand et al., 2015; Pöschko et al., 2017). In the current report, we focus on transient phenomena occurring when SN spectra are measured after and during applied pulsed field gradients. Our findings prove that the magnitudes of SN peaks depend on the immediate gradient history. The experimental evidence appears to support a paradigm of "refocused spin noise", which we call a spin-noise gradient echo (SNGE). Sequences composed of two or three magnetic field gradients in the absence of RF pulses can be used to obtain information on the transverse relaxation rates of protons and the diffusive mobility of molecules in pure liquids and their mixtures.

The interference between weak gradients and radiation damping in spin-noise spectra has been discussed previously (Pöschko et al., 2017). In the present report, we restrict experiments and discussions to a regime, where the combined homogeneous and inhomogeneous transverse relaxation rate exceeds the radiation damping rate. This allows us to use a relatively simple model based on assuming random RF excitations as the source of spin noise. The original spin-noise imaging (SNI) experiments were performed at similar conditions (Müller and Jerschow, 2006). The experimental SN spectra described here were recorded within this particular regime which is characterized by positive spin-noise signals, i.e. noise levels at nuclear spin resonance frequencies which exceed the Nyquist-Johnson circuit noise power level in the absence of spins (Nausner et al., 2009; Desvaux, 2013; Ferrand et al., 2015; Jurkiewicz, 2015).

## 2 Results and Discussion

To observe the described phenomena a cryogenically cooled high resolution liquids NMR probe optimized for $^1$H detection is used to acquire short noise blocks (i.e. short acquisition periods) without any prior RF excitation, but in the presence of and/or preceded by linear magnetic field gradients aligned along the static magnetic field axis ($z$). The noise blocks are then Fourier transformed to yield power spectra, which are co-added. The $z$-gradients are chosen sufficiently strong in order to observe an increase of noise power at the nuclear spin resonances and to avoid the non-linear distortions observed for weaker gradients (i.e. for gradient broadening smaller than the resonance line width) (Ferrand et al., 2015; Pöschko et al., 2017). Thus, the gradient strength during acquisition is set to induce sufficient spectral broadening to quench radiation damping effectively while still allowing for chemical shift discrimination. Comparing spin-noise power spectra recorded with a gradient applied during acquisition only, to spectra with an additional gradient (of same sign and amplitude) applied before acquisition reveal that the latter are of slightly higher spectral amplitude than the former. This intensity enhancement is even more pronounced if the two-gradient experimental schemes shown in Fig. 1(a) are compared: the sign of the pre-acquisition gradient $G_1$ is inverted between two separate two-gradient experiments. Afterwards, a SN power spectrum of the experiment with positive $G_1$ gradient is subtracted from the SN power spectrum of the experiment with negative $G_1$ gradient.

The SN power spectra of the experiments labelled $I(+)$ and $I(-)$ in Fig. 1(b) differ in intensity by about 20%. This observation is rather puzzling since such noise power signals are proportional to the spectral density over this frequency range, and in the time-domain proportional to the amount of autocorrelation. This means that an increase of the autocorrelated "component" during acquisition could be explained by refocusing of that component. Simulations, using Wolfram Mathematica™ (Wolfram Research Inc., 2012), help to further illustrate this phenomenon (see Supporting Information). The chosen experimental condition for these simulations corresponds to that where an increase of spin noise is observed. Hence, the model we use presumes that SN originates from a series of small random excitation events, each of random timing, random phase and random small flip angle. In this simple model we neglect effects by radiation damping because the inhomogeneous broadening by the gradients exceeds the resonance linewidth, i.e. the radiation damping rate, and Bloch equations applicable. Therefore, spin noise contributes additively to the other Johnson-Nyquist noise sources: the resonance circuit, the transmission line and the preamplifier. This simplification is further justified by simulations using an extended Bloch-equation model (Schlagnitweit et

al., 2012) for the simulation of small flip angle spectra, as detailed in the Supporting Information. Fig. S1 shows that in the presence of gradients of opposite sign (Fig. 1, case $I(-)$) and in the presence of transverse relaxation, an "incoherent echo" appears, with its center at the time where $\frac{\delta_2}{\delta_1} \approx 0.5$. These simulations indicate that, even if random processes without phase coherence are involved, the capabilities of defocusing and refocusing individual coherences by field gradient pulses are preserved.

The noise power amplitude of the $I(+)$ experiments (Fig. 1) is due to incoherent excitation occurring only during the second gradient. In contrast, the additional contributions observed in the $I(-)$ experiments result from excitation events occurring during the first gradient $G_1$ that are refocused by the second gradient $G_2$.

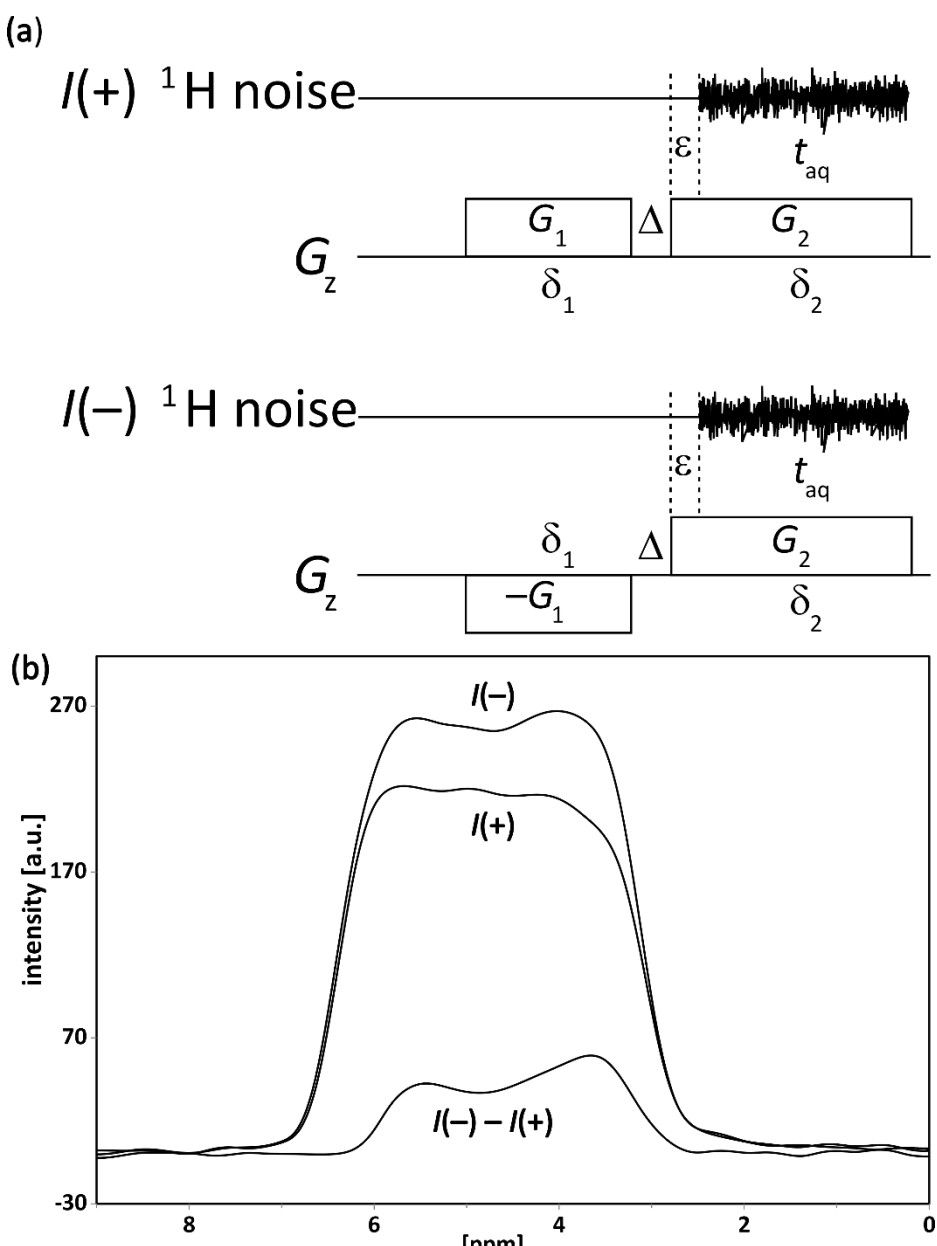

Figure 1: (a) Two-gradient pulse sequence (notably devoid of any RF pulses) used to demonstrate the principle of the spin noise gradient echo (SNGE). The noise blocks of the $I(+)$ and $I(-)$ experiments are stored separately and processed as described elsewhere (Nausner et al., 2009; Pöschko et al., 2017). The power spectra of $I(+)$ and $I(-)$ shown in panel (b) were used to calculate a difference spectrum $I(-) - I(+)$. Experimental parameters were: $G_1 = G_2 = 3.2\ \frac{mT}{m}$, $\delta_1 = 2\ ms$, $\Delta = 0.1\ ms$, gradient stabilization delay $\epsilon = 0.07\ ms$, acquisition time $t_{aq} = 3.69\ ms$. 2048 noise blocks were Fourier transformed and their power spectra added for each profile. The line shapes in panel (b) should ideally be rectangular gradient profiles. Deviations are caused by the non-ideality of the gradient system, the finite sample limits and residual radiation damping.

As in common RF pulsed gradient-echo experiments with two gradients (Tanner and Stejskal, 1968), the delay $\Delta$ in the sequence of Fig. 1(a) can be varied resulting in changes of amplitudes for the respective experiments. The differences between

the integrals of these $I(+)$ and $I(-)$ $z$-profiles, i.e. the spin-noise gradient echo (SNGE) amplitudes, decrease due to transverse relaxation and molecular displacement. Thus, a quantitative measure of the apparent transverse relaxation time $T_2^*$ can be extracted from the variation of the integrated SNGE difference spectra as a function of the delay $\Delta$. Fig. 2 illustrates the results of such a process. The extracted $T_2^*$ values include contributions from the spin-spin- relaxation time, instrumental broadening, residual radiation damping, and displacement along the $z$-axis.

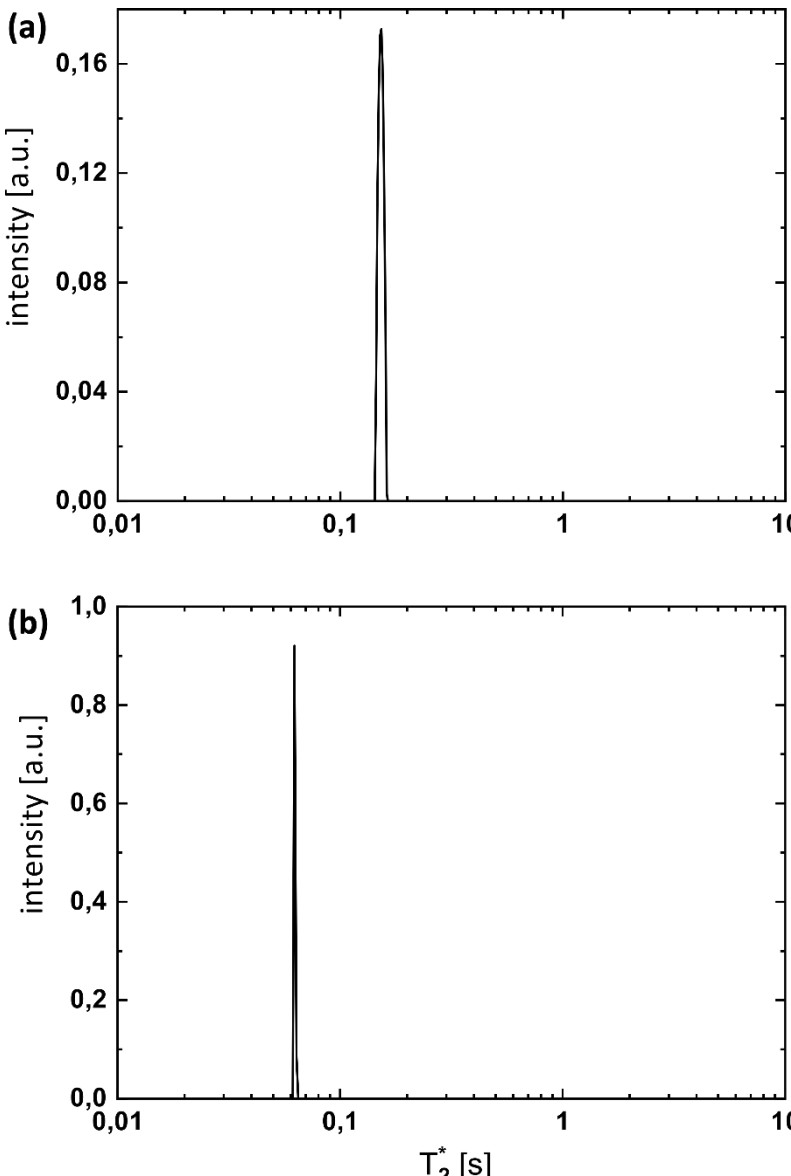


Figure 2. $T_2^*$ distributions obtained from inverse Laplace transform (ILT) (as implemented in MATLAB) (MATLAB, 2010) of the data sets measured in the $I(+)$ and $I(-)$ $^{1}$H SNGE experiments for different delays $\Delta$. (a) obtained from the $^{1}$H signal of $H_2O$ in 90%:10% $H_2O$:$D_2O$ (relaxation constant is about $160\,\mathrm{ms}$), (b) from the $^{1}$H signal of 90% acetone with 10% acetone-d6 (relaxation constant is about $62\,\mathrm{ms}$). $\Delta$ varied from 0 to $70\,\mathrm{ms}$ at constant parameters of: $G_1 = 1.26\frac{\mathrm{mT}}{\mathrm{m}}$, $G_2 = 10.7\frac{\mathrm{mT}}{\mathrm{m}}$, $\delta_1 = 35\,\mathrm{ms}$, gradient
stabilization delay $\epsilon = 0.07\,\mathrm{ms}$, acquisition time $t_{aq} = 3.69\,\mathrm{ms}$. Thus, with the ILT analysis, it is possible to extract a distribution of relaxation components $f(T_2)$. (Berman et al., 2013; Rodin, 2018) In particular, for pure liquids, this distribution $f(T_2)$ showed one relaxation peak.

For a common spin-echo experiment the overall time domain signal, $M(t)$ can be modelled by Eq. (1) (Rodin, 2018):

$M_t = \int_0^\infty \mathrm{e}^{-\frac{t}{T_2}} f(T_2) \mathrm{d}T_2$           (1)

This describes a superposition of individual signals relaxing independently at their respective decay rates $T_2$. The signal components are weighted by a function $f(T_2)$ which allows to discriminate between the different relaxation rates affecting

$M(t)$. The mathematical structure of Eq. (1) is that of a Laplace transform and hence $f(T_2)$ can be determined from $M(t)$ by applying the inverse transform (ILT).

If SNGE data as a function of $\Delta$ can be modelled by Eq. (1) (a function of $t$) analysis by an inverse Laplace transform is allowed. Relaxation constants can be determined in an alternative approach by fitting the normalized differences between $I(-)$ and $I(+)$ ¹H SNGE intensities as a function of the delay $\Delta$ with a single exponential function $e^{-\Delta/T_2^*}$. Here we find that both methods result in the same relaxation constants.

To separate the contribution of transverse relaxation to the decay of the spin-noise signal from the contribution by diffusion, one can exploit the fact that the decay by diffusion depends on the gradient amplitude while decay by transverse relaxation does not. The $z$-profile changes (different widths) owed to variations in gradient amplitude $G_2$ prevent a simple adaptation of the scheme in Fig. 1(a). Therefore, we introduce an improved experiment, which uses three gradient pulses. It allows one to keep the gradient during acquisition (third one) and hence the $z$-profile width constant, while the amplitude of the first and 130 second gradient are varied (Fig. 3).

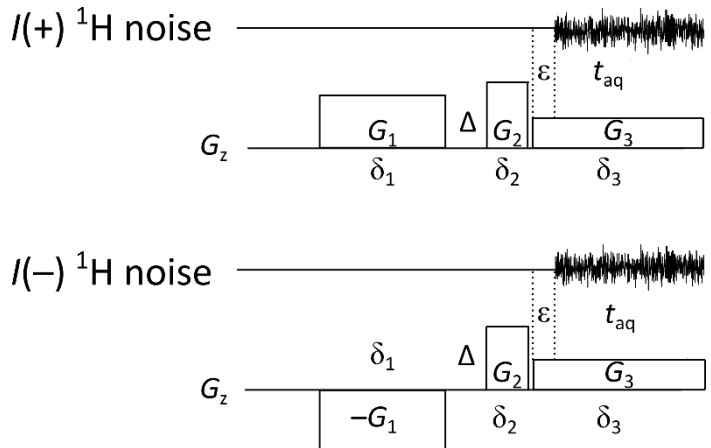

**Figure 3. Three-gradient spin noise detected diffusion experiment. The gradient ratios are adjusted according to Eq. (2). Gradient $G_2$ and and $G_3$ are separated by a short switching delay of $2 - 5\mu s$. Typical values in three-gradient sequences for diffusion (e.g. Fig. 5) and relaxation (e.g. Fig. 6) experiments are: $\delta_1 = 2ms$, $\Delta = 0.1ms$, $\delta_2 = 1ms$, gradient stabilization delay $\epsilon = 0.05$ ms,** 135 **acquisition time $t_{aq} = 3.69ms$). Acquisition is running during 3rd gradient pulse**

Apart from providing $z$-profiles of equal width, this acquisition scheme offers additional advantages as the maximum of the noise gradient echo can be adjusted to occur in the center of the acquisition time. Assuming a weak effect of transverse relaxation, this can be achieved by adjusting the gradient ratios according to Eq. (2) which derives from simulations results 140 shown in Fig. S1:

$$G_2 = \frac{G_1\delta_1 - G_3\delta_3}{\delta_2} \tag{2}$$

Typical NMR diffusion experiments apply RF and gradient pulses and exploit the dependence of the generated gradient echoes signals on gradient amplitudes for the determination of diffusion coefficients (Tanner and Stejskal, 1968; Rodin, 2018). In the SNGE experiment (Fig. 3 and Fig. 4), amplitudes $G_1$ and $G_2$ are incremented systematically, while exceeding the broadening 145 by radiation damping but allowing for separation of individual chemical shifts. (Fig 4)

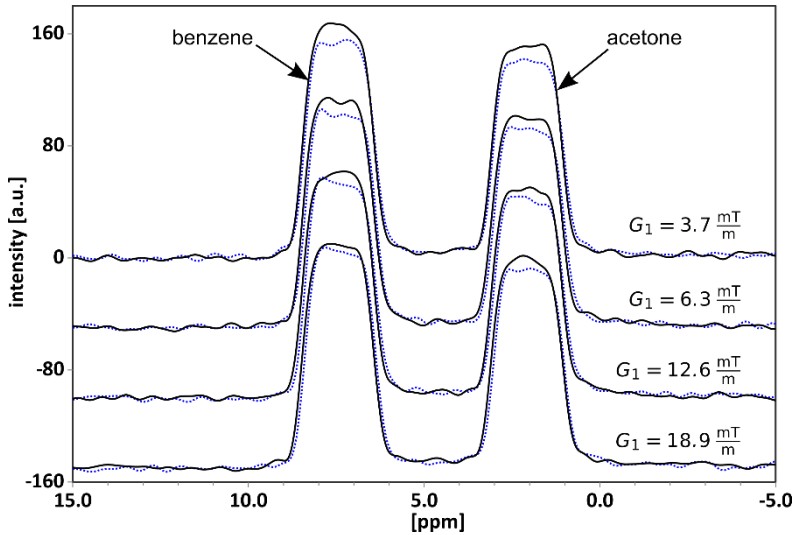

**Figure 4. Results from four SNGE experiments on a 1:1 mixture of acetone and benzene (with 10% of acetone-d6 for locking) recorded according to the three gradient sequence of Fig. 3 using the indicated $G_1$ gradient amplitudes at a constant $G_3$ of $1.6 \frac{mT}{m}$ while adjusting $G_2$ according to Eq. (2). The respective lower traces correspond to the $I(+)$ sub-experiment, the higher ones are of the $I(-)$ measurements. SNGE spectra are presented for four $G_1$ gradient values, increasing from top to bottom: $3.7\frac{mT}{m}$, $6.3\frac{mT}{m}$, $12.6\frac{mT}{m}$, $18.9\frac{mT}{m}$. ($\delta_1 = 2$ ms, $\Delta = 0.015$ ms, $\delta_2 = 1$ ms, gradient stabilization delay $\epsilon = 0.05$ ms, acquisition time $t_{aq} = 3.69$ ms). Chemical shifts: $2.1$ ppm (acetone); $7.2$ ppm (benzene).**

Quantitative measurement of diffusion coefficients $D_i$ requires normalization of the diffusion experiment spectra. For SNGE diffusion experiments one cannot resort to the zero-gradient experiment normalization as in pulsed diffusion NMR (Tanner and Stejskal, 1968; Hrabe et al., 2007; Kuchel et al., 2012; Berman et al., 2013; Rodin, 2018). Instead, we use the maximum difference signal $[I(-) - I(+)]_{max}$ as observed at the smallest $G_1$ gradient strength used for each peak. Assuming the validity of Eq. (2), a SNGE attenuation can be defined as:

$$\ln \frac{I(-) - I(+)}{[I(-) - I(+)]_{max}} = -D_i\gamma^2(G_2\delta_2 + G_3\delta_3)^2 \left(\Delta + \frac{2\delta_1}{3}\right)$$

$$= -D_i\gamma^2(G_1\delta_1)^2 \left(\Delta + \frac{2\delta_1}{3}\right) \tag{3}$$

Where $\gamma$ is the gyromagnetic ration. Eq. (3) shows a linear dependence of the SNGE attenuation on $G_1^2$ and a direct proportionality between the ratio of slope of the SNGE attenuation and the diffusion coefficient $D_i$.

Fig. 5 shows the result of applying the three-gradient pulse scheme to a $H_2O:D_2O$ solution (90%:10%) (Another example based on a mixture of acetone and benzene is shown in Fig. 4 and Fig. S2). Analysis of the dependence of the SNGE attenuation $\ln \frac{I(-)-I(+)}{[I(-)-I(+)]_{max}}$ on $G_1^2$ indicates a reasonable linear behavior as predicted by Eq. (3) except for very small $G_1^2$ values.

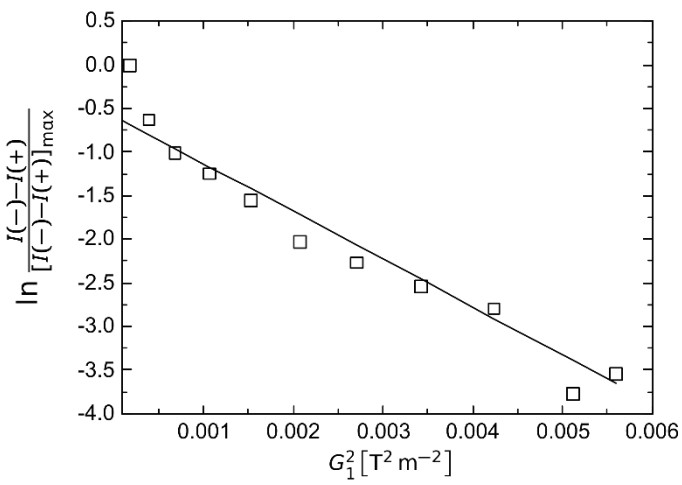

**Figure 5.** $G_1^2$-dependence of the normalized difference-spectra integrals of the $z$-profiles in SNGE $I(-)$ and $I(+)$ diffusion experiments from the three-gradient pulse scheme (Fig. 3) applied to 90%:10% H₂O:D₂O at $T = 295$ K. The solid line is added for guidance for the eye to emphasize the approximately linear part of the attenuation curve. ($G_2 = G_1$, $G_3 = 1.6\frac{mT}{m}$, $\delta_1 = 2$ ms, $\Delta = 0.1$ ms, $\delta_2 = 1$ ms, gradient stabilization delay $\epsilon = 0.05$ ms, acquisition time $t_{aq} = 3.69$ ms).

For very small gradient amplitudes, the assumption that the gradient broadening is larger than the radiation damping rate fails and non-linear behavior due to radiation-damping feedback fields have to be taken into account. SN experiments in the presence of weak gradients display for example spectral "hole burning" effects (Pöschko et al., 2017). This effect was simulated using a modified Bloch equation approach (Bloom, 1957; Schlagnitweit et al., 2012) emulating gradients by slices of different $B_0$ fields, all linked through a feedback field as generated by one and the same RF coil (Pöschko et al., 2017). Comparison

between experiments and such theoretical simulations were in very good agreement (Pöschko et al., 2017).

    This particular phenomenon can explain the discrepancies between Eq. (3) and the experimental data for very small $G_1^2$ values in Fig. 5. For our setup, z-profiles without any of these disturbances were observed for gradient amplitudes exceeding roughly $2\frac{mT}{m}$. Hence, restraining the data collection/analysis to the interval where $G_1 > 2\frac{mT}{m}$, is the most suitable experimental condition at which the SNGE sequence can be applied for the purpose of diffusion coefficient determination.

In a SNGE diffusion experiment on a 1:1 mixture of acetone and benzene, the SNGE attenuation vs $G_1^2$ (Fig. S2) allowed a rough estimation of the ratio of slope for the acetone and benzene component in this mixture separately. The ratio of these two slopes is comparable with the value derived from classical pulse-field gradient (PFG) spin echo measurements. From this one can conclude that, in two-component mixtures with sufficiently different chemical shifts, it is possible to observe SNGE attenuation and to extract separate SNGE attenuation curves for each component. A qualitative comparison of how SNGE

signals are attenuated in diffusion experiments of mixtures of components differing in both chemical shifts and diffusion coefficients appears feasible.

    A severe limitation of inducing SNGE attenuation by $G_1$ variation in the three-gradient scheme stems from gradient limits. In order to obtain a measurable decrease in echo intensity, data must be acquired over a wide enough $G_1$ range and this range must reside in the range where diffusion dominates the SNGE attenuation (linear $G_1^2$ dependence, see Fig. S2 top). In our case,

reliable SNGE experiment were not possible for gradient strengths higher than $75\frac{mT}{m}$, due to power limits on the fast gradient duty cycle. In order to circumvent this hardware limitation, an alternative implementation of the three-gradient pulse sequence is considered, where constant gradient amplitudes are used but the delay $\Delta$ between $G_1$ and $G_2$ is varied. For the 1:1 mixture of benzene and acetone, $G_3$ is chosen such that chemical shift discrimination is possible and no distortion on the sample profiles was observed (Fig. S3).

In Fig. 6 we report the SNGE attenuation for a mixture of acetone and benzene as a function of the delay $\Delta$. Increasing $\Delta$ causes a readily observable SNGE attenuation. However, the decay is the combined effect of simultaneous relaxation and diffusion. Both solvents display similar transverse relaxation rates $T_2^*$ in this mixture and hence, the difference of the two slopes can be interpreted as the difference in diffusion coefficients of the two, with the faster diffusion for acetone and the slower for benzene. For a systematic separation of relaxation and diffusion contributions SNGE attenuation curves for different

gradient amplitudes $G_1$ can be acquired and analyzed simultaneously (See Fig. S4).

    The apparent transverse relaxation times derived from this three-gradient experiment are approximately 10 times longer than the ones that can be extracted from directly recorded spin noise spectra by line shape analysis (which by the Wiener-Khintchine theorem (Wiener 1930; Khintchine 1934) is equivalent to an autocorrelation analysis). This is expected because radiation damping is heavily affected by feedback from the pre-amplifier circuit (Pöschko et al., 2017). In the hardware used the

preamplifier is actively detached from the rf receiver circuit by impedance switching, as long as no acquisition is running. Therefore, when tuning and matching are set up for the SNTO (Marion and Desvaux, 2008; Nausner et al., 2009; Pöschko et al., 2014) as in the experiments reported here, the radiation damping rate is a maximum during acquisition and at a much lower value before and after acquisition. The apparent transverse relaxation rate thus is reduced in the period $\Delta$ due to the tuning

offset occurring when the preamplifier is on high impedance, which partially quenches radiation damping. The influence of
preamplifier feedback on spin noise has been described and simulated in detail in reference (Pöschko et al., 2017). Observation
of relaxation phenomena during the indirect evolution period in the three-gradient experiment also opens the possibility to
probe the spectral density which lies at the root of the spin-noise phenomenon (Field and Bain, 2010; Field and Bain 2013;
Field 2014) under conditions which are not dictated by the particular implementation of the receiver and preamplifier circuitry
(see supporting information)

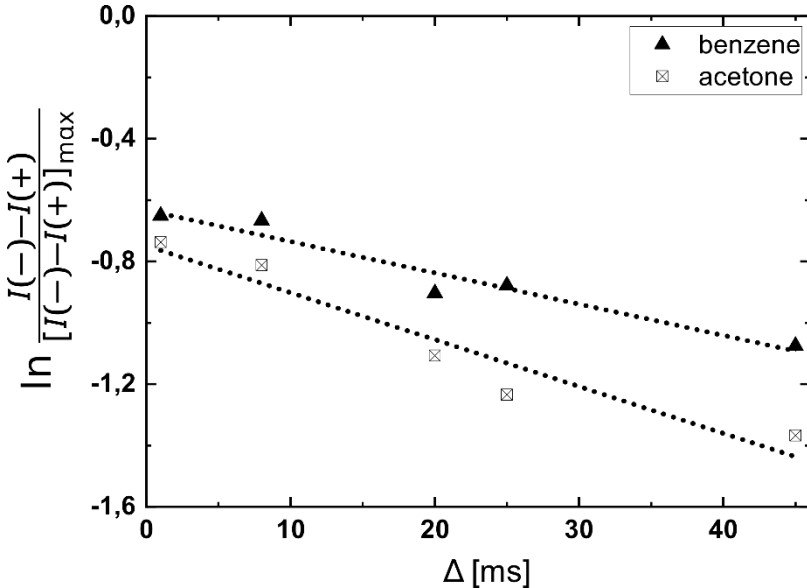

**Figure 6. $^1$H SNGE attenuation experiments in a 1:1 acetone-benzene mixture for different delays $\Delta$ between $G_1$ and $G_2$ using the three-gradient sequence. (top) data set (triangles) are for the benzene component (chemical shift 7.2 ppm, $T_2^* = 91$ ms to 95 ms), (bottom) data set for the acetone component (chemical shift 2.1 ppm, $T_2^* = 60$ ms to 65 ms). $\Delta$ varied from 0 ms to 45 ms at constant parameters of: $G_1 = 31.5\frac{mT}{m}$, $G_2 = 60.1\frac{mT}{m}$, $G_3 = 1.6\frac{mT}{m}$. ($\delta_1 = 2$ ms, $\delta_2 = 1$ ms, gradient stabilization delay $\epsilon = 0.05$ ms, acquisition time $t_{aq} = 3.69$ ms). The respective SNGE spectra at $\Delta = 8$ ms, 20 ms conditions are shown in Fig. S3.**

## 3 Conclusions

In this work, we report first explorations of diffusion and relaxation characterizations based on nuclear spin-noise detection. In the SNGE (Spin-Noise Gradient Echo) acquisition scheme, no RF excitation is used, and the NMR signals are retrieved as autocorrelation functions (by way of Fourier transformation) of the acquired noise data. In order to encode the effect of transverse relaxation and/or diffusion pulsed field gradients are applied prior to the noise acquisition. This gradient encoding alters the autocorrelation functions of the acquired noise data, if a refocusing gradient is simultaneously active during the noise acquisition. The relative signs and amplitudes of these gradients induce spin-noise amplitude changes from which information on transverse relaxation and diffusion can be, at least semi-quantitatively, extracted.

Analysis of the influence of the chosen experimental conditions on the SNGE phenomenon allows one to identify several advantages and limitations in terms of gradient sequence implementation and hardware parameters. Firstly, since no RF excitation is used for detecting magnetic resonance, SNGE seems particularly attractive for spin systems exhibiting long longitudinal relaxation times ($T_1$) as no recycle delay is needed. For example, in diamond $T_1$ times of nearly 100 h (!) have been found (Reynhardt and Terblanche, 1997). Our group has also demonstrated the utility of spin noise measurements at low temperatures (Pöschko et al., 2015; Pöschko et al., 2016). where spin lattice relaxation can also be extremely slow, while $T_2^*$ is short.

Also, the simultaneous determination of transverse relaxation rates and diffusion coefficients is attractive for mixture characterization in porous media. Additionally, in situ oil well exploration, (Prammer, 2004) or other applications of NMR in confined spaces would profit from the miniaturization and simplification possible by a detection-only electronic setup. However, the first implementations of the SNGE have also revealed constraints provided by the hardware and the range of experimental systems, which can be studied. Firstly, since SNGE is based on spin-noise measurements, the coupling between the magnetization and the detection circuit should be sufficiently strong to induce radiation damping (the fact that RF excitation is not needed enlarge the detection-circuit designs) (Ferrand et al., 2015). Secondly, and more important for SNGE, the range of useful pulse field gradient amplitudes as provided by common NMR spectrometers is very limited. For very small gradient amplitudes, profile distortions appear, preventing the signal analysis for diffusion characterization, and the extraction of reference measurements in the absence of gradients. Also, with the high-resolution probe we used, resorting to very high gradients was impossible, because of duty cycle and power restraints and also because detection must be done in the presence of a gradient. We, nevertheless, show that some routes exist, without adapting the hardware for circumventing these amplitude-gradient issues: for instance, by using a three-gradient pulse scheme instead of the two-gradient echo, by changing the intergradient echo time and the gradient amplitudes or by performing comparative experiments between two solvents. For allowing this direct comparison, we show that gradient strengths lower than the chemical shift separation allow the obtaining of separate profile images for different spin isochromats, from which we extract apparent transverse relaxation time $T_2^*$ for components in SNGE experiments largely independent of radiation damping and compare molecular diffusion coefficients.

## 4 Experimental

Spin noise data was collected as described in elsewhere (Nausner et al., 2009) typically using a total of $2^{16}$ noise acquisition blocks. Some variations in the number of noise blocks were depending on concentration and type of experiment. The individual noise blocks were stored separately, then Fourier transformed, converted to power spectra and co-added.

Solvents (acetone, benzene, DMSO) were supplied from Sigma-Aldrich in analytical grade. In the experiments with pure solvents 5–10% of the respective deuterated solvent (Sigma-Aldrich) were added for field-frequency locking. In the experiments run on the mixtures of acetone and benzene, the field-frequency lock signal

was provided by acetone-d6. All raw data used in the main text have been collected and processed using a Bruker 700 MHz Avance III NMR spectrometer equipped with a 5 mm TCI cryo-probe (manufactured in 2011) with an internal z-gradient coil (max. gradient $0.625\frac{\text{T}}{\text{m}}$) and Bruker Topspin 3.2 (Topspin, 2012) NMR software.

Temperature of the samples was controlled to 295 K, the RF coil temperature was $20.1 \pm 0.1$ K. The probe was tuned to the spin-noise tuning optimum (Nausner et al., 2009; Pöschko et al., 2014). Noise blocks (the equivalent to FIDs in pulsed NMR) were recorded in the presence of gradients, with an inhomogeneous broadening exceeding the radiation damping rate by at least a factor of two. Such magnetic field gradients are commonly used to alleviate the effects radiation damping in high-resolution NMR (Henry, 1986). The maximum pulsed $B_0$ field gradients applied were also relatively weak causing a line broadening of a few kHz in the proton NMR spectra. For rapid parameter adjustments during setup (e.g. positioning of the echo, gradient shapes and pre-emphasis) small flip angle excitation experiments were used to circumvent the time requirements of non-optimized SN experiments.

## 5 Abbreviations

FID        free induction decay;

NMR     nuclear magnetic resonance;

NQR     nuclear quadrupole resonance;

SN        spin noise;

PFG       pulsed field gradient;

280 RD    radiation damping;

RF        radio frequency;

SNTO      Spin Noise Tuning Optimum;

SNGE      spin noise gradient echo;

SNI       spin noise imaging;

## 285 6 Data availability

The data in the figures are available at https://doi.org/XXX (Rodin et al., 2020.). The software used for simulations is available upon request

**Supplement link (will be included by Copernicus)**

The supplement related to this article is available online at: https://doi.org/XXX

## 290 Author contribution

VVR, SJG and MB did the measurements. VVR did all the simulations and analysis. NM and HD supervised the work. The paper was written by all authors with final editing by MB and NM.

## Competing interests

The authors declare that they have no conflict of interest.

## 295 Acknowledgements

The authors are grateful to Dr. Gaspard Huber for the exchange visits and help during this project.

## Financial support

This work was supported by the Austrian Science Funds (FWF) and the Agence Nationale de Recherche (ANR) in the joint project IMAGINE (FWF project number I1115-N19, ANR project 12-IS04-0006) and by the EGIDE-ÖAD AMADEUS 300 Austrian French exchange programme (no 28948WD and FR 11/2013).

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
