# Peer review of "Spin-Noise Gradient Echoes"

_Magnetic Resonance, 2021_

## Author Comment (AC1)

The manuscript "Spin-Noise Gradient Echoes" is the continuation of authors research in the area of spin noise and its applications. Authors are presenting interesting new application of spin noise into the study of spin diffusion showing that diffusion coefficients can be estimated without applying of radio frequency pulses. The beauty of creation the spin noise shows lack of necessity of brute force in the form of pulses because coherence is created spontaneously by statistic of spin fluctuations. Therefore, in their experiment necessary coherence for studying spin diffusion is coming not from pulses but from natural spontaneous fluctuations of spin noise system.

In the course of the manuscript I can see several problems that I would like to address. On Figure 1(b) there is lack of axis label and also lack of corresponding units. Existing number are very small and hard to read. Looking at the same Figure 1 (a) one can see that duration of gradients are not the same. δ2 is longer than δ1. However, in the figure caption says they are the same δ1 = δ2 = 2 ms. This shows lack of consistency. This also suggests that acquisition time t is 2ms long which seems to be improbable. This requires clarification.

On the same Figure the delay Δ = 0. 015 ms. Which seems to be very short. How the ring down of the gradient coil behaves during this 15 microsecond delay? And what are the consequences of this interference? Clarification is needed.

I could not find a definition in the caption or text for the meaning of the delay ε = 0. 007 ms.

The Figure 1S.- is showing the numerical simulation of spin noise according to parameters: ð® ð = −ðð; ð® ð = ðð; ð¸ = ð; ð ð = ð; ð = ð; ð» ð = ðð. One would like to ask what means T2=10? Is this microseconds, miliseconds, seconds? T2 always has dimension and without such is meaningless. The same question applies to the rest of the listed parameters. Why can not put real physical numbers into simulation? Lack of real number in simulation indicates that we are talking not about physical reality but something that exists in abstraction or virtual reality. On the other hand spin noise is a physical reality and needs to be treated with real numbers. This issue requires clarification. Also assumption "spin noise can be envisaged as a series of spontaneous excitation events of a random phase " is raising question " is this reflecting physical reality or spin noise abstraction? To reconcile this, the work of T. R. Field and A.D. Bain, Appl. Magn. Reson. 38, 167 (2010) and T.R. Field and A.D. Bain, Physic. Rev. E 87(2), 022110-1 (2013) about properties of spin noise and its origin needs to be discussed and cited.

Figure 2.- lack of labels and units on both axis.

Figure 3. -It could beneficial for reader if captions contains values of δ1, δ2, Δ and ε for every spectrum recorded according to this scheme.

Figure 4. -lack of x axis label and units. Numbers are small and hard to read.

Equation 3. -Symbol "γ" is not defined.

Figure 5. -Y axis label in wrong place.

Figure 2S a and b. -the denominator's subscript for y axis label is not readable.

Authors bring several times the magnitude of radiation damping as important parameter. However, they did not present the actual number. I think the value of radiation damping and spin-spin relaxation time will give better view of the physical reality. I would like to suggest to consider measurement of radiation damping and give this to reader.

In equations S1,S3,S4 transverse magnetization is represented by symbol "m0". In NMR tradition which goes back to Bloch, the magnetization used to be expressed by capitol :"M, Mx, My, Mz" I would suggest considering to keep up with common tradition.

The manuscript has the potential for important contribution to the application and understanding of spin noise echo. However, I can see significant number of problems that needs to be addressed before can be published. I recommend for publication after major revision is made.

Reply

**Citation**: https://doi.org/10.5194/mr-2021-55-RC1
[Figure]

My manuscript overview

---

## Author Response (AR1)

Magn. Reson. Discuss., referee comment RC1
https://doi.org/10.5194/mr-2021-55-RC1, 2021

[Figure]

**Comment on mr-2021-55 – Response by Authors**

Anonymous Referee #1
* * *
Referee comment on "Spin-Noise Gradient Echoes" by Victor V. Rodin et al., Magn. Reson. Discuss., https://doi.org/10.5194/mr-2021-55-RC1, 2021

Lines numbers are from:
SNGE_Reply-to-Referee-highlight-changes.pdf
SNGE-SI_Reply-to-Referee-highlight-changes.pdf

In the course of the manuscript I can see several problems that I would like to address. On Figure 1(b) there is lack of axis label and also lack of corresponding units. Existing number are very small and hard to read.

Fixed;
We reformatted all figures

L90,L105, L131,L143,LL162,L214, L214
L29SI, L52SI, L75SI, L85SI

Looking at the same Figure 1 (a) one can see that duration of gradients are not the same. $\delta 2$ is longer than $\delta 1$. However, in the figure caption says they are the same $\delta 1 = \delta 2 = 2$ ms. This shows lack of consistency. This also suggests that acquisition time t is 2ms long which seems to be improbable. This requires clarification.

The durations of $\delta 2$ / $t_{aq}$ are short when compared to those in typical spin noise and canonical NMR experiments, where they are controlled by the length of $T^*_2$. In our paper the spin noise signal is recorded in the presence of the gradient $G_2$, causing chemical shift dispersion and hence a shortening of the echo signal decay time. The overall maximum of the echo signals is expected to appear at times shorter than the dephasing time $\delta 1$, since dephasing starts with each spin noise processes happening during $\delta 1$, and hence, are also refocused earlier than in canonical gradient echo experiments where the echo maximum appears at times $\delta 2 = \delta 1$.

Fixed; We added the explicit parameter values for of the experiments as used for Fig 1(b). Adding gradient stabilization delay $\epsilon$ and acquisition time $t_{aq}$
L94,L95

On the same Figure the delay $\Delta = 0. 015$ ms. Which seems to be very short.

Fixed, typo to $\Delta = 0.1$ ms,
L94

How the ring down of the gradient coil behaves during this 15 microsecond delay? And what are the consequences of this interference? Clarification is needed.

Since this is a typical value used in canonical gradient echo experiments, we do not expect / observe detrimental effects due to gradient ring-down of G1.

We, however, do observe / expect interference effects for small gradient amplitudes $G_1$ of about 0.8-1 mT/m. These effects have been discussed in (Pöschko et al., 2017 … gradients which cause a line broadening equal to or less than the radiation damping rate, a complex line shape is observed …) The ring down of the gradient coil by G2 is taken care of by a gradient stabilization delay $\varepsilon=0.07$ms (see next comment)

I could not find a definition in the caption or text for the meaning of the delay $\varepsilon = 0.007$ ms.

Fixed; Annotation of the gradient stabilization delay $\varepsilon$ has been added to the figure caption of Fig 1. The typo was corrected to $\varepsilon = 0.07$ ms
L94, L95

The Figure 1S.- is showing the numerical simulation of spin noise according to parameters: G1 = -10; G2=10; Y=1; m0=1; l=1; T2=10. One would like to ask what means T2=10? Is this microseconds, miliseconds, seconds? T2 always has dimension and without such is meaningless.

The same question applies to the rest of the listed parameters. Why can not put real physical numbers into simulation? Lack of real number in simulation indicates that we are talking not about physical reality but something that exists in abstraction or virtual reality. On the other hand spin noise is a physical reality and needs to be treated with real numbers. This issue requires clarification.

Fixed; The simulations in Fig S1 have been rerun using the experimental parameters ("physical numbers") of the experiment in Fig 1(b). The simulations use a stochastic coherent excitation model for spin noise signals in the presence of pulsed gradients. This model is used to estimate the optimum duration of the refocusing gradient during acquisition. As mentioned in the paper, we are well aware of situations where this model has to fail and we do not claim that this very simplistic model is even the best possible model at the experimental conditions of the current article. However, our focus is the experimental demonstration that spin-noise gradient echoes can be measure and that they can be exploited for diffusion measurements.
L29S and caption

Also assumption "spin noise can be envisaged as a series of spontaneous excitation events of a random phase " is raising question " is this reflecting physical reality or spin noise abstraction? To reconcile this, the work of T. R. Field and A.D. Bain, Appl. Magn. Reson. 38, 167 (2010) and T.R. Field and A.D. Bain, Physic. Rev. E 87(2), 022110-1 (2013) about properties of spin noise and its origin needs to be discussed and cited.

We do not present/intent the stochastic coherent excitation model to be physical. But rather see it as a kind of quick estimate (Fermi estimate) of the signal extremal points in SNGE experiments. The simulations are added intentionally in the SI only and should be seen more as a thought experiment.

We are grateful for the two suggested papers, as they point the way to analyzing the spin-noise dynamics in SNGE experiments in a fundamental way. We added the references in the SI part of the article. Together with a short mention in the SI text.
L46Sf

Figure 2.- lack of labels and units on both axis.

Fixed;
reformatted Figure
Deleted some floating text: … $e^{-\Delta/T_2^*}$ Y-scale is an intensity in arbitrary units…
L105, L112

Figure 3. -It could beneficial for reader if captions contains values of δ1, δ2, Δ and ε for every spectrum recorded according to this scheme.

Fixed; we added missing experimental parameters to each of the figure captions of article and SI L95, L110, L133, L148, L165f, L218f, L32Sff, L79S

Figure 4. -lack of x axis label and units. Numbers are small and hard to read.

Fixed; reformatted figure
L 143

Equation 3.-Symbol "γ" is not defined.

Fixed: γ is gyromagnetic ratio

L157

Figure 5. -Y axis label in wrong place.

Fixed; reformatted figure
L163

Figure 2S a and b. -the denominator's subscript for y axis label is not readable.

Fixed; reformatted figure
L53S

Authors bring several times the magnitude of radiation damping as important parameter. However, they did not present the actual number. I think the value of radiation damping and spin-spin relaxation time will give better view of the physical reality. I would like to suggest to consider measurement of radiation damping and give this to reader.

We know from previous experiments and simulations (Pöschko et al., 2017) that experiments, using gradients 1 mT/m or larger, are not affected by the radiation damping. Further, accurate measurement of radiation damping rates is not a trivial task, and we are in the process of assembling an article on this very topic.

In equations S1,S3,S4 transverse magnetization is represented by symbol "m0". In NMR tradition which goes back to Bloch, the magnetization used to be expressed by capitol :"M, Mx, My, Mz" I would suggest considering to keep up with common tradition.

Fixed; changed all m0 to M0
L14S,L17S, L21S, L23S,L 25S,L27S,L28S,L32S,L34S

The manuscript has the potential for important contribution to the application and understanding of spin noise echo. However, I can see significant number of problems that needs to be addressed before can be published. I recommend for publication after major revision is made.

Lines numbers are from:

SNGE_Reply-to-Referee-highlight-changes.pdf

SNGE-SI_Reply-to-Referee-highlight-changes.pdf

1-I think the authors could elaborate a bit more in the main text on the mechanism of spin-noise echo formation. In particular, I wonder if a description that takes into account the spin noise spectral density (or, equivalently, the spin noise correlation function) isn't appropriate. Along the same lines, it would be good if the authors could include a correlation curve (as determined from the spin noise time traces in the absence of gradients) and show how this compares to values derived from the gradient protocol (where decoherence and diffusion can be determined separately).

*The echo formation is a dynamic process. Our view of the mechanism is deliberately simplistic, assuming random creation of coherence in different locations (voxels) of the sample. The computation of the autocorrelation function is equivalent to the Fourier transformed power spectrum of the (spin) noise by the Wiener-Khinchin relation. Therefore, the line width in the noise power spectrum obtained without gradients also is determined by $T_2$\*.*

*We have added a paragraph preceding the conclusions referring to the difference of directly detected spin noise line shapes and the apparent transverse relaxation occurring in the intergradient delay in our three-gradient experiment. It is the difference in radiation damping caused by pre-amplifier switching, which has the main influence on T2\*. Therefore, computational separation of relaxation and diffusion is not straightforward and would require access to probe parameters, which are beyond user control. There is a new Figure in the Supporting Information, comparing the directly recorded spin noise spectra to the line widths obtained from the gradient experiment.*

*L198-L210 Discussion of direct detected SN line shape and under SNGE conditions.*

*L209: added references; (Field and Bain 2010, Field and Bain 2013 and Field 2014)*

*L90S-L95S Discussion of T2\* in SNGE experiments and no-gradient experiments*

*L96S Figure S5 and caption was added*

2-The impact of radiation damping on spin noise detection is mentioned a couple times but the main ingredients are described very superficially. With the understanding that the subject has been previously discussed elsewhere, it would

be a good idea to devote a paragraph or two to better highlight the central ideas, even if briefly.

*In the presence of a weak static magnetic field gradient, i.e. one which causes a broadening less than the one caused by radiation damping one observes a bump in the NMR-signal. This is illustrated in the supplementary information of Ref. (Poeschko 2017). Repeating this in the current paper would, in our opinion, distract from the main subject as it cannot easily be summarized in a paragraph or two and would even require additional figures. We have however modified the quotation to point directly to the supplementary information of the relevant paper, as the matter is not discussed in the main text of that reference (Pöschko 2017).*

*(no changes made)*

3-In the concluding section, the authors highlight some of the areas where the present technique could find use, including the case where the system T1 is long. While this is true on general grounds, I wonder how long should the T1 be to make this technique competitive. I think a ballpark numerical assessment is in order here. I recommend the same for other directions the authors identify as promising.

*The T1 advantage could be relevant in samples whereT1 is in the order of hours to days, as for example in ultra-cool samples (Pöschko 2015, Pöschko 2016) or dilute spins in crystalline solids like diamond. We have added two sentences in the conclusions pertaining to this request.*
*L231-234 added comment on long T1 experiments*
*L232 added reference (Reynard and Terblanche, 1997)*
*L233 added reference (Pöschko 2016)*

4-I had a hard time interpreting some of the figures. In particular, I found several occasions where all axis labels and units are missing and numerical labels are too small to be seen (if at all present). I strongly suggest the authors should more carefully review all figures to ensure they convey all necessary information.

*We have improved all figures and added units and axis labels where they were missing.*

---

## Author Response (AR2)

Response to Editor's request

**MR-2021-55**

*However, I believe some minor changes are still missing in the caption of figure 3 (values of delta1, delta2, delta3 and Delta), which should fixed before publication.*

Response:

The Figure caption has been updated.

In addition, we have added two references which were accidentally omitted in the previous version.  (see pp. 7, 11, 12)